# Effect of pH, Reducing Sugars, and Protein on Roasted Sunflower Seed Aroma Volatiles

**DOI:** 10.3390/foods12224155

**Published:** 2023-11-17

**Authors:** Jessica Laemont, Sheryl Barringer

**Affiliations:** Department of Food Science and Technology, The Ohio State University, 110 Parker Food Science and Technology Building, 2015 Fyffe Road, Columbus, OH 43210, USA; laemont.1@osu.edu

**Keywords:** sunflower seeds, Maillard reaction, SIFT-MS, roasting, volatiles

## Abstract

Sunflower seeds are a popular snack in many countries, such as the United States, China, and Spain. Sunflower seeds are typically roasted to create desirable aromas before being eaten. The desirable aromas are created by the Maillard and lipid oxidation reactions. Increasing the volatiles created by these reactions can create a more desirable product, increasing consumer acceptance of sunflower seeds. Seeds were soaked in solutions at pH 4, 7, and 9 and with added glucose, fructose, whey protein isolate, or whey protein concentrate before roasting. The resulting seeds were evaluated by selected-ion flow tube mass spectrometry to determine the volatile concentrations and by an untrained panel of consumers to determine acceptability. Increasing the pH increased the pyrazines but did not affect other volatiles. Adding reducing sugars or whey protein increased most volatiles. The fructose increased dimethylpyrazines, 2-methylpyrazine, and trimethylpyrazine concentrations more than glucose. However, the glucose increased furfural concentration more than fructose. The whey protein concentrate increased volatile levels more than any other treatment. The total Maillard volatiles and Browning index were increased by the same treatments. Sensory indicated that fructose increased desirable aroma the most, followed by whey protein treatments, and both were liked more than the pH 7 control. Optimizing roasting conditions by increasing the pH and reducing sugar and protein content can favor the Maillard reaction conditions, increasing the positive aromas associated with roasted sunflower seeds.

## 1. Introduction

Sunflowers are an important agronomic crop in the USA. There are two types of sunflower seeds, oil seeds and confectionary seeds. Confectionary sunflower seeds are a popular snack, especially in baseball, as sunflower seeds are eaten instead of chewing tobacco [1]. These seeds are an important source of edible oil and protein. The protein content of the seeds ranges from 20 to 30% [2] and the lipid content ranges from 48 to 53% [3]. Sunflower seeds are eaten roasted, which creates their desirable aroma.

The roasting of sunflower seeds is necessary to create their desirable aroma. The sensory properties (color, odor, hardness, flavor, and total acceptance) all improve significantly after roasting [4]. Several studies have focused on determining the optimum roasting conditions for sunflower seeds [5,6]. Sunflower seeds treated with protein were found to have greater flavor, caramel aroma, umami characteristics, and overall acceptability than sunflower seeds without [7]; thus, sunflower seed acceptability can be improved with the correct treatments.

Alpha-pinene, 2,5-dimethylpyrazine, furfural, hexanal, and phenylacetaldehyde were reported to be the dominant compounds in roasted sunflower seeds in one study [8], while alpha-pinene, benzaldehyde, 2,5-dimethylpyrazine, furfural, hexanal, methional, and phenylacetaldehyde were reported to be the dominant compounds in another study [9]. The most important contributors to the characteristic aroma of roasted sunflower seed oil are the dimethylpyrazines [10]. Most of the aroma in roasted sunflower seeds is created by the Maillard reaction, with important contributions from lipid oxidation.

The Maillard reaction is affected by pH, reducing sugar, and protein type and concentration [11]. A higher pH facilitates the formation of pyrazines [12]. The initial condensation reaction is initiated by nucleophilic attack of the nitrogen, which is highly dependent on pH. The reaction is faster under alkaline conditions because the amino group is not protonated; thus, Amadori and Heyns products undergo 2,3-enolization generating 1-deoxyhexosones, which proceed to produce pyrazines [13]. When the pH is below the pka of the amino group, this initial step of the Maillard reaction is slower due to reduced nucleophilicity, which increases reactivity with the carbonyl groups of reducing carbohydrates and favors the formation of furfurals and acid-catalyzed sugar degradation [14]. The greatest variety of pyrazine compounds, especially trimethylpyrazine, 2,5-dimethylpyrazine, tetramethylpyrazine, pyrazine, and 2-methylpyrazine, is produced at approximately pH 9 [15]. On the other hand, furfural formation increases in acidic conditions [16,17,18].

The specific amino acid is important in determining the pyrazines formed since the nitrogen atom in the pyrazines comes from the amino compounds [13]. Heating whey protein produces high concentrations of pyrazines, including 2,5-dimethylpyrazine and 2-methylpyrazine [19].

Different reducing sugars exhibit different reactivities, with fructose being more reactive than glucose [14]. Thus, fructose typically produces more 2-methylpyrazine and dimethylpyrazines than glucose [20]. Fructose also increased furfural formation in baked sponge cake more than glucose [21]. Furfural formation is lower with hexose sugars because of the higher energy requirement for carbon-carbon bond cleavage [17].

Lipid oxidation is often associated with rancidity and product deterioration. However, it also produces desirable aromas which are critical components of foods, such as potato chips and baked goods [22]. The difference between a desirable and undesirable aroma is the relative proportion and concentration of the volatile oxidation products. The presence of other roasted notes from the Maillard reaction and the aromas produced from lipid oxidation signal to the consumer a roasted rather than a rancid product [22]. High temperatures increase lipid oxidation because the unsaturated fatty acids react more quickly at high temperatures producing volatile oxidation products [23].

Hexanal, pentanal, and pentanol increase during the roasting of sunflower seeds, especially at higher temperatures and longer roasting times [8]. However, the concentration of 1-octen-3-ol decreases with time and temperature [8]. 2-Heptanal, 1-octen-3-ol, and hexanal contribute to the rancid odor of roasted sunflower seeds [24].

There is conflicting literature on the effects of reducing sugars on lipid oxidation volatile formation. Fructose and glucose accelerate the autoxidation of linoleic acid in an aqueous emulsion [25]. However, other studies found that reducing sugars reduced lipid oxidation. Crackers made with glucose had delayed hexanal formation during storage [26], and heated fructose exhibited an antioxidant effect on minced fish [27].

The pH, type, and amount of reducing sugars and protein affect both the Maillard reaction and lipid oxidation and thus, the volatiles associated with the desirable aromas of roasted sunflower seeds. The objective of this study was to improve roasting conditions to maximize positive volatile aroma compounds from sunflower seeds by modifying pH, reducing sugar content, and protein content.

## 2. Materials and Methods

### 2.1. Roasting Methods

#### 2.1.1. Pretreatments

Panther sunflower seeds were deshelled to obtain the kernel. A 30 g portion of the kernel was used for each treatment. The control consisted of kernels soaked for 15 h at 25 °C in 70 mL of deionized water at pH 7. Treatments consisted of kernels soaked in 70 mL of phosphate buffer at pH 4, 70 mL of phosphate buffer at pH 9, 70 mL of 20% glucose (Sigma Aldrich^®^, St. Louis, MO, USA), 70 mL of 20% fructose (NOW Foods, Bloomingdale, IL, USA), 70 mL of 17% whey protein isolate (Mill Haven Foods, New Lisbon, WI, USA), or 70 mL of 17% whey protein concentrate (Mill Haven Foods). The sugar and protein solutions were prepared on a weight-by-volume basis. The pH meter (Orion 2 Star, Thermo Fisher Scientific, Waltham, MA, USA) was calibrated with standards at pH 4, 7 and 10. It was rinsed with distilled water between measurements of the buffer solutions.

#### 2.1.2. Roasting

After soaking, the sunflower seeds were spread in a monolayer and dried for 24 h at 25 °C. The layer of seeds was then roasted in an oven (Blue M Stabil-Therm, Blue M Electric Company, Blue Island, IL, USA) at 165 °C for 8 min.

### 2.2. Headspace Volatile Quantification

#### 2.2.1. Sample Preparation

Each replicate consisted of 10 g of kernels ground for 10 s (Magic Bullet Express, Homeland Housewares, New York, NY, USA). To determine volatiles in the treatments themselves, 10 g of the treatment solution was used. Each sample was placed into a 500 mL Pyrex bottle and equilibrated in a 50 °C water bath (Thermo Precision 2864 Circulating Water Bath, Thermo Fisher Scientific, Waltham, MA, USA) for 90 min [28]. The bottles were capped with an open-top septum lined cap for analysis. Head space analysis was conducted in triplicate.

#### 2.2.2. SIFT-MS

The headspace volatile compounds were analyzed using a selected-ion flow tube mass spectrometer (SIFT-MS) (SYFT Voice200ultra, Syft Ltd., Christchurch, New Zealand). SIFT-MS is a direct mass spectrometric method that analyses volatile organic compounds in air with typical detection limits at parts-per trillion by volume level. Real-time, quantitative analysis is achieved by applying precisely controlled soft chemical ionization. Chemical ionization of the headspace volatiles occurs by their reaction with selected precursor ions (H_3_O^+^, NO^+^, or O_2_^+^) generated from a microwave discharge ion source [29]. The concentration of a volatile compound in the headspace is equal to the measured product ion count rate times the instrument calibration factor, divided by the predetermined rate constant (k) for that volatile times the reagent ion concentration [30,31]. During analysis, all samples were run using selected ion mode (SIM) scans. Quantification of the volatile compound concentrations was determined using the reaction rate coefficients presented in Table 1 [32]. Calibration was performed to quantify linearity, range, accuracy, precision, and limit of detection using a pressurized mixture of certified gas standards (benzene, ethylbenzene, toluene, m-xylene, o-xylene, and p-xylene), with each having a concentration of 2 ppm (±5%) in nitrogen (Air Liquide America Specialty Gases LLC, Plumsteadville, PA, USA) and regulated to a pressure of 21 kPa (3 psi). Ultra-high-purity helium (UHP 99.999%; Praxair, Columbus, OH, USA), regulated to a pressure of 170 kPa (25 psi) was used to thermalize reagent ions and act as the carrier gas. The headspace was sampled by piercing the septa of a 500 mL glass bottle cap with a 14-gauge passivated needle, which was directly affixed to the sampling inlet port of the V200. The sample inlet flow was normalized to a rate of 0.346 ± 0.014 TorrL s^−1^ or 26 ± 1 sccm (standard cubic centimeters per minute) under standard state ambient temperature (298 K) and pressure (1 bar). For each reagent ion (H_3_O^+^, NO^+^, and O_2_^+^), a 50 ms time limit per scan was allowed with 4 cycles/repeats per scan for a total sample run time of 60 s. The high-performance inlet/heated inlet extension (HEX) temperature was maintained at 175 ± 1 °C to prevent any condensation along the sample line and to ensure complete volatilization of compounds from the sample headspace to the instrument reaction tube. Maintaining this temperature along the HEX line helps to clear out the compounds adhering to the surface of the sampling line and thus decreases the probability of cross interference (carryover effects) between samplings. Heated HPLC water (as blank) and multiple air scans were analyzed between the samples to also minimize carry-over effects.

Some volatiles can produce the same mass/charge (*m*/*z*) value causing conflicts and have been reported as a mixture. Dimethylpyrazine isomers are a mixture of 2,3-dimethylpyrazine, 2,5-dimethylpyrazine, and 2,6-dimethylpyrazine. 2-Pentylfuran isomers are a mixture of 2-pentylfuran and 2-ethylpyrazine. Pinene isomers are a mixture of alpha-pinene and beta-pinene. 3-Methyl-2-ethylpyrazine isomers are a mixture of 3-methyl-2-ethylpyrazine and 2-ethyl-3-methylpyrazine.

### 2.3. Color

Each replicate for color analysis consisted of a 1 g portion of ground sunflower seeds placed in a small transparent plastic bag to prevent damage to the colorimeter lens. Color measurements were performed at ambient temperature using a Color Quest XE colorimeter (Hunter Associate Laboratory, Inc., Reston, VA, USA) The colorimeter measures the reflectance of light from the surface of the sample, from which it calculates the *L** *a** *b** values of the sample. Using the *L** *a** *b** values, the Browning index (*BI*) was calculated to determine the intensity of the brown color. Color analysis was performed in triplicate.
BI=([100(x−0.31)])/0.17
$$x=(a^{*}+1.75L^{*})/(5.645L^{*}+a^{*}-3.012b^{*}))$$

### 2.4. Sensory Analysis

Based on recommended study specifications, 105 untrained subjects were recruited for the sensory study. The panelists comprised students and staff at Ohio State University. Panelists were screened to ensure they did not have an allergy to sunflower seeds, were free from known gustatory, severe vision, or olfactory deficits and had refrained from smoking for at least two hours prior to the start of the experiment.

Samples consisted of 10 g ground roasted sunflower kernels and were evaluated at room temperature in 100 mL Pyrex sample bottles with a lid and wrapped in aluminum foil in order to prevent the color of the sample from affecting the panelist’s feedback. The experiments were conducted with a within-subjects design wherein each panelist served as his/her own control. Samples were coded with 3-digit random blinding codes.

The samples were pH 7 (control), fructose, whey protein isolate, and whey protein concentrate-treated roasted sunflower seeds. Panelists were asked to smell the samples and assess their liking of the samples. Liking of the aroma was assessed on a 9-point scale from 9 = like extremely to 1 = dislike extremely. Panelists were also asked to rank the samples from 1 = most desirable aroma to 4 = least desirable aroma.

This study was reviewed and approved by the Ohio State University Institutional Review Board and informed consent was obtained from each subject prior to their participation in Study Number 2023E0409.

### 2.5. Statistical Analysis

Data were analyzed with JMP version 17 (Cary, NC, USA). One-way analysis of variance (ANOVA) and Tuckey’s range test was performed on volatile concentrations and sensory data.

## 3. Results and Discussion

### 3.1. Effect of Roasting on Maillard Reaction Volatiles

Roasting is known to improve desirable aromas and increase the sensory acceptability of sunflower seeds [4,8], with the most desirable aroma compounds formed from the Maillard reaction. Kernels were soaked in pretreatments consisting of pH 4, pH 7 (control), pH 9, glucose, fructose, whey protein isolate, or whey protein concentrate to increase desirable aromas that occur during roasting. The volatiles in the treatments alone were measured, and, in most cases, the volatiles were at very low levels (Table A1). All of the volatiles were significantly lower in the treatments than in the raw sunflower seeds, with the exception of the lipid oxidation generated volatiles hexanal and pentanal, which were present in the whey protein concentrate and whey protein isolate at levels of up to 25 ppb. For all treatments, roasting significantly increased the formation of the Maillard reaction volatiles, including 2-methylpyrazine, dimethylpyrazine isomers, trimethylpyrazine, tetramethylpyrazine, furfural, benzaldehyde, phenylacetaldehyde, and methional, contributing to the desirable aromas of roasted sunflower seeds (Table 2). Other studies also found that roasting generated these Maillard reaction volatile compounds in sunflower seeds [7,8,9,12,24] and in roasted almonds [33], walnuts [34], and pumpkin seeds [28].

2,5-Dimethylpyrazine was previously reported to be the second most abundant volatile in roasted sunflower seeds after alpha-pinene [8]. Among the Maillard volatiles, we also found the dimethylpyrazine isomers to be the most abundant for the roasted sunflower seed treatments (Table 2). 2,5-Dimethylpyrazine has a nutty/roasty aroma and was above its odor threshold of 100 ppb [8] for many of the treatments. Therefore, it contributes to the pleasant odor of the roasted kernels.

Tetramethylpyrazine is another Maillard volatile important to the roasted aroma and was also found in roasted pumpkin seeds [28]. Tetramethylpyrazine contributes a nutty/earthy aroma and was above its odor threshold of 38 ppb [35] for many of the treatments. Therefore, it also contributes to the pleasant odor of the roasted kernels.

2-Methylpyrazine and trimethylpyrazine are Maillard reaction volatiles important to the roasted aroma and were also found in roasted almonds [33]. Both 2-methylpyrazine and trimethylpyrazine contribute a nutty aroma. While 2-methylpyrazine and trimethylpyrazine increased for many treatments, neither exceeded their odor threshold of 30,000 and 730 ppb, respectively [36].

Furfural is an early Maillard reaction product formed during the degradation of sugars and was also found during the roasting of almonds [33]. Furfural contributes a caramel-like aroma. While furfural increased for many of the treatments, it was not over its odor threshold of 3000 ppb [37].

The Strecker aldehydes benzaldehyde, phenylacetaldehyde, and methional contribute almond/sugary, honey-like, and cooked potato aromas, respectively. Benzaldehyde, phenylacetaldehyde, and methional were above their odor thresholds of 60 [38], 22 ppb [8], and 0.05 ppb [8], respectively, for many of the treatments.

#### 3.1.1. Effect of pH on Maillard Reaction Volatiles in Roasted Sunflower Seeds

The kernels were soaked in solutions of pH 4, 7, and 9. The pH influences the pathway of the Maillard reaction, changing the concentrations of the volatiles that are produced. Increasing the pH from 4 to 9 significantly increased the formation of the pyrazines 2-methylpyrazine, dimethylpyrazine isomers, trimethylpyrazine, and tetramethylpyrazine (Figure 1). The nucleophilic attack of the amino group during the initial stage of the Maillard reaction favors a basic pH; thus, a higher pH increases the formation of pyrazine compounds during roasting [13]. Previous studies have found similar results. In a model system, adjusting the pH to 8 facilitated the formation of pyrazine compounds during roasting [12]. 2,5-Dimethylpyrazine, trimethylpyrazine, and tetramethylpyrazine increased in roasted cocoa beans when the pH was increased from 5.2 to 8 [39].

Increasing the pH to pH 9 increased pyrazine compound formation, thus increasing the concentration of desirable roasted aromas. However, the concentration of the Strecker aldehydes benzaldehyde, phenylacetaldehyde, and methional remained constant (Figure 1). The final stage of the Maillard reaction is the Strecker degradation process, which produces aldehydes and aminoketones [22]. The aminoketone further reacts to produce pyrazine compounds [22]. At a more basic pH, the reaction shifts to produce more pyrazines [13]; thus, we expected more Strecker aldehydes to be formed as well.

Furfural was also not affected by a change in pH (Figure 1). It was expected that furfural would increase at pH 4 because low pH increases the reactivity of the carbonyl groups of reducing carbohydrates during the Maillard reaction and favors the formation of furfural [14].

#### 3.1.2. Effect of Reducing Sugars on Maillard Reaction Volatiles in Roasted Sunflower Seeds

The kernels were soaked in glucose and fructose solutions. The sugar in sunflower seeds is predominately sucrose, at 3.2 g/100 g [40], which gradually breaks down into fructose and glucose during roasting [41]. Sunflower seeds have only 3% sugar compared to 20–30% protein [2], which indicates that the reducing sugar is the limiting reagent. Increasing the reducing sugar content by increasing the amount of glucose or fructose in sunflower seeds was expected to increase the desirable aromas generated from the Maillard reaction. The reducing sugars glucose and fructose significantly increased the Maillard reaction volatiles 2-methylpyrazine, dimethylpyrazine isomers, trimethylpyrazine, tetramethylpyrazine, and furfural (Figure 1). However, the addition of the reducing sugars did not affect the concentration of benzaldehyde, methional, and phenylacetaldehyde, which remained constant (Figure 1). As expected, adding reducing sugars significantly increased furfural formation because furfural is formed from the thermal degradation of sugars (Figure 1).

As well as the amount, the type of reducing sugar is also important to the Maillard reaction. Reducing sugars have different reactivities that affect the formation of the desirable aroma volatiles. Fructose increased the concentrations of the pyrazines more than glucose (Figure 1). Fructose is more reactive than glucose [14]. Similarly, in a lysine solution, fructose also increased 2,5-dimethylpyrazine and 2-methylpyrazine more than glucose [41].

However, the results were the opposite for furfural, and glucose increased furfural concentration more than fructose (Figure 1). The literature is conflicting on the effects of reducing sugar type on furfural formation. In sponge cake, glucose produced significantly more furfural than fructose [21]. However, in a fruit juice system, the reaction was temperature dependent, with glucose producing more furfural at lower temperatures and fructose producing more furfural at higher temperatures [42].

Adding reducing sugars increased the pyrazine and furfural concentrations more than changing the pH affected the concentration of these volatiles. Neither affected the concentration of the Strecker aldehydes. Thus, the addition of reducing sugars increases the desirable aromas more than increasing pH.

#### 3.1.3. Effect of Protein on Maillard Reaction Volatiles in Roasted Sunflower Seeds

The addition of Maillard peptides to boiled sunflower seeds was shown to increase their volatile concentrations and sensory acceptability [7]. Thus, adding whey proteins to sunflower seeds should increase the desirable aroma volatiles formed during the Maillard reaction. The addition of whey protein concentrate and whey protein isolate increased the formation of the pyrazines (Figure 1). Pyrazines are formed from the amino compounds in proteins which are an important factor in determining the backbone of pyrazine structure [13]. Therefore, increasing the protein content is expected to increase the pyrazine compound formation, increasing the desirable aromas from the Maillard reaction.

While the pH and sugar content did not affect Strecker aldehyde formation, adding whey protein increased the concentration of the Strecker aldehydes benzaldehyde, phenylacetaldehyde, and methional. The addition of the free amino acids in the whey protein increases the Strecker aldehydes, and thus, Strecker aldehydes are formed from the reaction between α-dicarbonyl and free amino acids after decarboxylation and deamination [43]. The addition of whey protein also increased furfural formation. Furfural was not expected to increase because it is a sugar degradation product that can form with or without amino acids during the early stage of the Maillard reaction [44]. However, whey protein concentrate and isolate contain lactose [45], a reducing sugar that can convert into furfural.

The type of whey protein solution had a significant effect on volatile formation as the whey protein concentrate produced higher volatile concentrations than the whey protein isolate for the pyrazines and Strecker aldehydes (Figure 1). The whey protein concentrate contained more lactose than the isolate. While the higher protein concentration in the isolate was expected to produce the most volatiles, the combination of additional protein with the higher concentration of reducing sugar, the limiting reagent, in whey protein concentrate likely caused the greater increase in volatiles.

Whey protein isolate and concentrate increased the pyrazines and Strecker aldehydes more than the reducing sugar and pH, thus increasing the desirable aromas. Furfural increased significantly only for the reducing sugar treatment.

### 3.2. Effect of Roasting on Lipid Oxidation Volatiles

Roasting significantly increased all lipid oxidation volatiles (Table 2). Lipid oxidation is often associated with rancidity and product deterioration, but it also produces desirable aromas, which are critical components of foods, such as potato chips and baked goods [22]. The difference between an undesirable and desirable aroma is in the relative proportion and concentration of the volatile oxidation products. In the case of desirable roasted aromas, the presence of other roasted notes from the Maillard reaction, along with the lipid oxidation aromas, signal a roasted product rather than a rancid product [22]. Some volatiles from lipid oxidation, such as 2-pentyl furan, have a fruity and earthy odor that can be desirable [9]. High temperatures increase lipid oxidation because the unsaturated fatty acids react more quickly at high temperatures [23].

Pentanal, 1-pentanol, and hexanal are formed from the lipid oxidation of linoleic acid, the most abundant fatty acid in sunflower seeds, at 32 g/100 g [34]. Pentanal is responsible for a fruity, nutty aroma, and was above its odor thresholds of 12 ppb in all treatments, therefore contributing to the pleasant aroma of the roasted kernels (Table 2) [46]. Pentanal was the most abundant volatile in roasted sunflower seeds suggesting that lipid oxidation is a key to the aroma of roasted sunflower seeds. Lipid oxidation was also crucial in creating a pleasant aroma in fragrant rapeseed oil, where the most prevalent type of volatiles were also aldehydes from the lipid oxidation of linoleic acid [23].

1-Pentanol contributes a sweet balsamic bready aroma. Pentanol was above its odor threshold of 470 ppb [8] for many of the treatments, contributing to the roasted kernels’ pleasant aroma. Pentanol increases with the time and temperature of roasting sunflower seeds [8].

Hexanal was not above its odor threshold of 479 ppb [8] for any of the treatments; thus, its fruity, fatty, rancid odor should not be detectable. Sunflower seeds that were roasted longer (15–60 min) than in this study reported hexanal at much higher levels and as the third most abundant volatile [8].

Alpha-pinene is a terpene found in raw sunflower seeds that increased during roasting. It contributes a pine aroma and was above its odor threshold of 6 ppb [8] in all treatments and the raw kernels, therefore contributing to the pleasant aroma of the kernels. Alpha-pinene was previously found to be the most abundant volatile in raw and roasted sunflower seeds [8].

#### 3.2.1. Effect of pH on Lipid Oxidation Volatiles in Roasted Sunflower Seeds

Kernels were soaked in pH 4–9 solutions. There was no significant effect of pH on pentanal, pentan-1-ol, and hexanal formation (Figure 2). It was not expected that pH would affect the formation of lipid oxidation volatiles.

#### 3.2.2. Effect of Reducing Sugars on Lipid Oxidation Volatiles in Roasted Sunflower Seeds

While there was no effect of pH, adding the reducing sugars glucose and fructose significantly increased the concentrations of pentanal, pentan-1-ol, and hexanal (Figure 2). In previous studies, fructose and glucose had contradictory effects on the formation of these volatiles. Fructose and glucose were found to accelerate the autoxidation of linoleic acid [25] but also to decrease hexanal formation in crackers [26] and serve as antioxidants in minced fish [27]. The type of reducing sugar (fructose versus glucose) did not affect lipid oxidation volatile formation.

#### 3.2.3. Effect of Whey Protein on Lipid Oxidation Volatiles in Roasted Sunflower Seeds

Adding whey protein concentrate and isolate significantly increased the formation of pentanal, pentan-1-ol, and hexanal (Figure 2). Whey protein concentrate and isolate contain 4 and 3.5% ash, respectively, which includes iron and copper that can catalyze lipid oxidation. Whey protein contains 25.7 and 2.6 mg/kg of iron and copper, respectively [47]. Iron and copper can accelerate oxidation when they undergo redox cycling to decompose hydroperoxides into reactive lipid radicals [48]. Iron and copper in the whey protein may have caused an increase in lipid oxidation volatile formation during roasting. The whey protein concentrate produced higher volatile concentrations than the whey protein isolate, possibly due to the higher ash content in the whey protein concentrate compared to the whey protein isolate.

### 3.3. Color

Increasing the pH, reducing sugar, or protein content was expected to increase browning because the Maillard reaction produces both volatiles and brown pigments [8]. The color of the samples was measured by colorimeter to determine the *L**, *a**, and *b** values, which were used to calculate the Browning index, ∆E values, and chroma. ∆E measures the total color difference, and chroma indicates the intensity of the color. The Browning index measures the purity of brown color, so when the Browning index value is high, there is more brown color [49].

The conditions that increased the total concentration of Maillard volatiles also increased the Browning index values (Figure 3). The chroma (Figure A1) and ∆E (Figure A2) values exhibited the same trend as the Browning index. In increasing order of total Maillard volatile concentration and brownness, the samples were raw, pH 4, pH 7, pH 9, whey protein isolate, glucose, fructose, and whey protein concentrate.

### 3.4. Sensory

Sensory studies were conducted on the most promising treatments, fructose, whey protein isolate, and whey protein concentrate and on the control samples (pH 7) to determine if soaking in these treatments prior to roasting increased liking among untrained consumers. The sensory results showed that untrained consumers can tell the difference in aroma between the roasted sunflower seeds soaked in water (control) and the roasted sunflower seeds soaked in whey protein or fructose (Figure 4). Both whey protein and fructose increased the desirable aromas. In increasing order of liking and ranking, the samples were pH 7, whey protein concentrate and whey protein isolate, and fructose. Previous studies also found that the addition of proteins improved the sensory acceptability of sunflower seeds [7].

The seeds soaked in fructose produced the highest liking score and were ranked the highest for the aroma of the roasted sunflower seeds. The fructose-treated seeds did not have the highest concentration of pyrazines or Strecker aldehydes, which could be expected to produce the most desirable odors. However, the fructose-treated seeds had the highest furfural concentration of all the samples (Figure 1). Furfural contributes a caramel-like aroma, which likely contributed significantly to the desirable aroma of the roasted sunflower seeds. The second most liked samples were whey protein isolate and whey protein concentrate. The whey protein samples had significantly higher pyrazine and Strecker aldehyde concentrations than the control pH 7 sample. The pyrazines and Strecker aldehydes contribute nutty, almond/sugary, cooked potato, and honey-like aromas, which may have increased liking.

Roasted sunflower seeds do not have a color standard like peanuts [50] or walnuts [51]. Peanuts must be a medium brown color but not lighter than USDA Color 2 nor darker than USDA Color 3 [50]. The color of walnuts is classified into one of the following categories: extra light, light, light amber, or amber, but not darker than the “amber” classification based on the USDA Walnut Color Chart [51]. Based on the liking of the fructose and whey protein roasted sunflower seeds, a Browning index value of 44–60 is a good starting point for developing a color standard for roasted sunflower seeds.

## 4. Conclusions

Both the Maillard reaction and lipid oxidation are important to create the desirable aroma profile of roasted sunflower seeds. The pyrazine concentrations were increased the most by the addition of whey protein, followed by reducing sugars, and then by increasing the pH to 9. Strecker aldehyde concentrations only significantly increased with the addition of whey protein. Furfural concentration only increased with the addition of reducing sugars. Lipid oxidation volatiles increased in concentration the most with the addition of whey protein and reducing sugars; however, pH had no effect. Sensory results indicate that consumers can tell the difference in the aroma between the roasted sunflower seeds. Fructose followed by whey protein treatments increased the desirable aromas the most and sensory indicated that the fructose and whey protein were liked more than the control. Treating with fructose or whey protein before roasting significantly enhanced the presence of desirable aromas and acceptability of roasted sunflower seeds.

## Figures and Tables

**Figure 1 foods-12-04155-f001:**
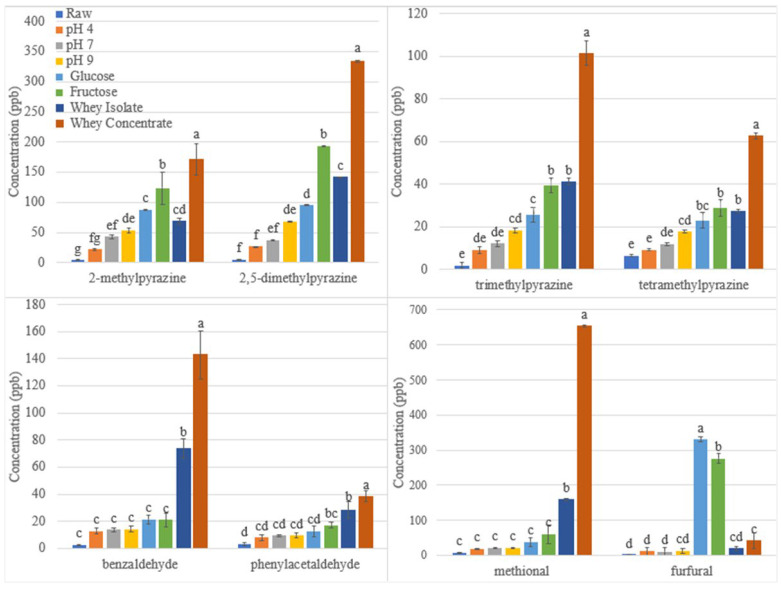
Formation of Maillard reaction volatiles in oven roasted sunflower seeds. Different letters for the same volatile indicate significant difference (*p* < 0.05).

**Figure 2 foods-12-04155-f002:**
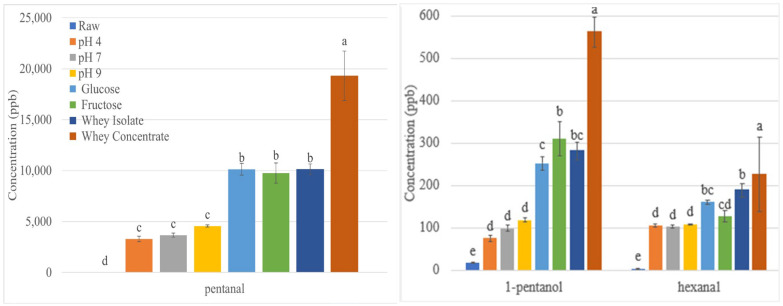
Formation of lipid oxidation volatiles in oven roasted sunflower seeds. Different letters for the same volatile indicate significant difference (*p* < 0.05).

**Figure 3 foods-12-04155-f003:**
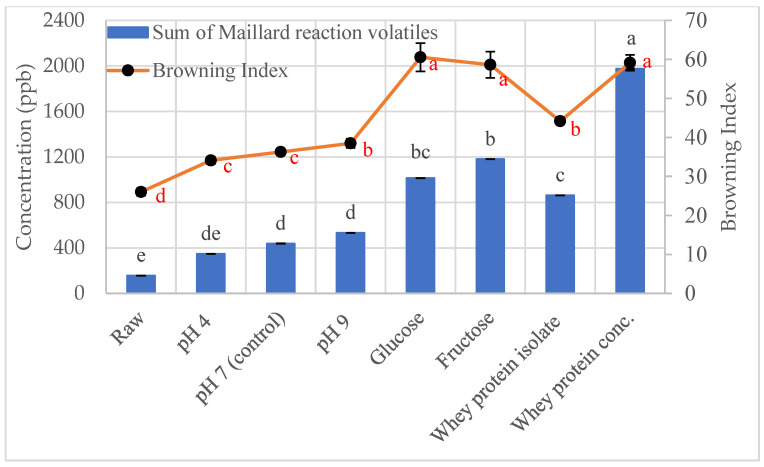
Sum of Maillard reaction generated volatiles (bars) compared to Browning index (points). The different letters in black indicate significant difference (*p* < 0.05) for the Maillard volatiles and the different letters in red indicate significant difference (*p* < 0.05) for the Browning index value.

**Figure 4 foods-12-04155-f004:**
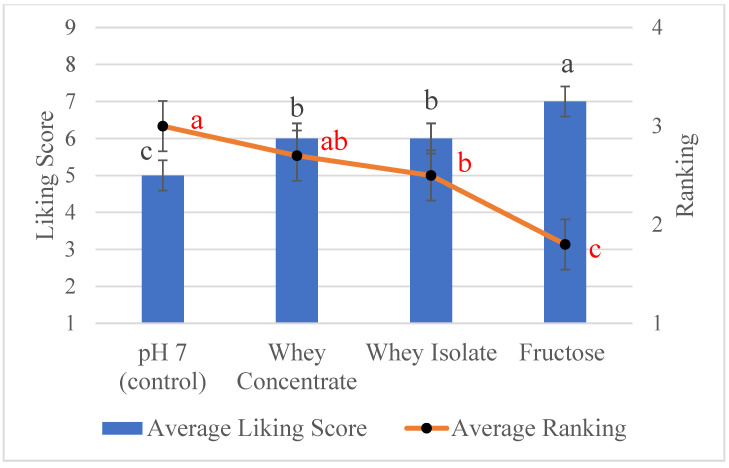
Average liking score (bars) and average ranking (points) of the aroma of roasted sunflower seeds treatments: pH 7, whey protein concentrate, whey protein isolate, and fructose. Different letters in black indicate significant difference (*p* < 0.05) in the liking score and different letters in red indicate significant difference (*p* < 0.05) for the ranking.

**Table 1 foods-12-04155-t001:** Detailed SIFT-MS Information of the Measured Volatile Compounds in Panther Sunflower Seeds.

Volatile	Reagent	Rate Constant (k) (10^−9^ cm^3^/s)	Mass (*m*/*z*)	Product
1,4-butyrolactone	NO^+^	3.5	116	C_4_H_6_O_2_.NO^+^
1-octanol	NO^+^	2.3	129	C_8_H_17_O^+^
1-pentanol	O_2_^+^	2.8	70	C_5_H_10_^+^
2,3-dimethyl-5-methylpyrazine	O_2_^+^	2.5	150	C_9_H_14_N_2_^+^
Dimethylpyrazine isomers	O_2_^+^	2.7	108	C_6_N_2_H_8_^+^
2-decenal	NO^+^	2.1	153	C_10_H_17_O^+^
2-heptenal	NO^+^	3.9	111	C_7_H_11_O^+^
2-isopropyl-3-methylpyrazine	H_3_O^+^	3.0	153	C_8_H_12_N_2_O.H^+^
2-methylpyrazine	NO^+^	2.8	94	C_5_H_6_N_2_^+^
2-pentylfuran isomers	NO^+^	2.0	138	C_9_H_14_O^+^
3-methyl-2-ethylpyrazine isomers	NO^+^	2.5	152	C_7_H_10_N_2_.NO^+^
Pinene isomers	NO^+^	2.3	136	C_10_H_16_^+^
benzaldehyde	O_2_^+^	2.4	106	C_7_H_6_O^+^
Decanal	NO^+^	3.3	155	C_10_H_19_O^+^
Furfural	NO^+^	3.2	96	C_5_H_4_O_2_^+^
Hexanal	NO^+^	2.5	99	C_6_H_11_O^+^
hydroxymethylfurfural (HMF)	O_2_^+^ NO^+^	2.5 2.5	126 126	C_6_H_6_O_3_^+^ C_6_H_6_O_3_^+^
Methional	O_2_^+^	2.5	104	C_4_H_8_O_5_^+^
Nonanal	NO^+^	2.7	141	C_9_H_17_O^+^
Octanal	NO^+^	3.0	127	C_8_H_15_O^+^
Pentanal	NO^+^	3.0	85	C_5_H_9_O^+^
phenylacetaldehyde	NO^+^	2.5	150	C_8_H_8_O.NO^+^
Pyrazine	NO^+^	2.8	80	C_4_H_4_N_2_^+^
tetramethylpyrazine	O_2_^+^	2.5	136	C_8_H12N^+^
trans-2-undecenal	H_3_O^+^	3.0	169	C_11_H_20_O.H^+^
Trimethylpyrazine	NO^+^ O_2_^+^	2.5 2.5	122 122	C_7_H_10_N_2_^+^ C_7_H_10_N_2_^+^

**Table 2 foods-12-04155-t002:** Average Concentration (ppb) of Volatiles in Roasted Sunflower Seeds for Each Treatment. Different letters for the same volatile indicate significant difference.

Volatile	Raw Seeds	pH 4	pH 7	pH 9	20% Glucose	20% Fructose	17% Whey Isolate	17% Whey Concentrate
1,4-butyrolactone	3.13 ^e^	15.1 ^de^	17.0 ^de^	23.9 ^d^	88.1 ^b^	104 ^a^	52.4 ^c^	104 ^ab^
1-octanol	1.64 ^d^	3.70 ^d^	4.46 ^d^	5.22 ^cd^	57.9 ^a^	55.3 ^a^	17.1 ^c^	41.6 ^b^
1-pentanol	17.9 ^e^	76.2 ^d^	98.7 ^d^	119 ^d^	252 ^c^	310 ^b^	282 ^bc^	562 ^a^
2,3-diethyl-5-methylpyrazine	0.60 ^e^	0.78 ^de^	1.33 ^cde^	1.31 ^cde^	2.10 ^c^	1.89 ^cd^	4.77 ^b^	7.89 ^a^
dimethylpyrazine isomers	4.88 ^f^	26.6 ^f^	37.4 ^ef^	68.2 ^de^	95.5 ^d^	193 ^b^	141 ^c^	333 ^a^
2-decenal	1.39 ^bc^	1.72 ^bc^	1.03 ^c^	1.38 ^bc^	5.20 ^a^	3.80 ^ab^	5.78 ^a^	6.03 ^a^
2-heptenal	0.86 ^e^	7.50 ^d^	8.26 ^d^	10.4 ^d^	19.3 ^c^	17.9 ^c^	25.7 ^b^	57.6 ^a^
2-isopropyl-3-methoxypyrazine	3.27 ^cde^	1.84 ^e^	2.08 ^e^	2.22 ^de^	5.21 ^c^	4.87 ^cd^	8.16 ^b^	13.5 ^a^
2-methylpyrazine	4.05 ^g^	22.3 ^fg^	43.0 ^ef^	53.6 ^de^	87.9 ^c^	124 ^b^	69.0 ^cd^	171 ^a^
2-pentylfuran	2.28 ^d^	3.65 ^cd^	3.69 ^cd^	4.92 ^cd^	7.06 ^bc^	10.4 ^b^	9.06 ^b^	21.9 ^a^
3-methyl-2-ethylpyrazine isomers	4.00 ^bc^	2.26 ^c^	2.55 ^c^	2.22 ^c^	5.17 ^bc^	6.46 ^b^	5.65 ^b^	11.2 ^a^
hydroxymethylfurfural (HMF)	1.98 ^d^	13.0 ^cd^	30.3 ^b^	18.8 ^bc^	21.6 ^bc^	22.4 ^bc^	22.6 ^bc^	56.0 ^a^
Pinene isomers	108 ^d^	191 ^c^	219 ^abc^	261 ^a^	242 ^ab^	264 ^a^	172 ^c^	204 ^bc^
benzaldehyde	2.49 ^c^	12.9 ^c^	13.7 ^c^	14.3 ^c^	21.4 ^c^	20.9 ^c^	73.7 ^b^	143 ^a^
decanal	2.04 ^c^	7.73 ^ab^	5.53 ^bc^	6.16 ^b^	8.46 ^ab^	6.94 ^b^	5.36 ^b^	11.5 ^a^
furfural	1.01 ^d^	11.4 ^d^	11.0 ^d^	12.4 ^cd^	333 ^a^	276 ^b^	19.8 ^cd^	41.2 ^c^
hexanal	4.54 ^e^	107 ^d^	103 ^d^	109 ^d^	161 ^bc^	127 ^cd^	189 ^b^	227 ^a^
methional	7.51 ^c^	19.2 ^c^	21.2 ^c^	20.7 ^c^	38.0 ^c^	60.4 ^c^	160 ^b^	653 ^a^
nonanal	2.67 ^f^	8.04 ^cde^	5.92 ^e^	6.20 ^de^	9.03 ^cd^	9.13 ^c^	11.8 ^b^	16.2 ^a^
octanal	1.96 ^d^	6.22 ^c^	5.91 ^cd^	5.66 ^cd^	20.1 ^a^	14.6 ^b^	8.51 ^c^	14.4 ^b^
pentanal	3.81 ^d^	3290 ^c^	3660 ^c^	4550 ^c^	10,140 ^b^	9760 ^b^	10,120 ^b^	19,300 ^a^
phenylacetaldehyde	3.37 ^d^	7.90 ^cd^	9.28 ^cd^	9.51 ^cd^	12.6 ^cd^	17.1 ^bc^	28.3 ^b^	38.4 ^a^
pyrazine	1.91 ^c^	2.38 ^c^	3.03 ^c^	3.61 ^c^	7.75 ^b^	6.99 ^b^	6.23 ^b^	14.4 ^a^
tetramethylpyrazine	6.52 ^e^	9.20 ^e^	11.8 ^de^	17.7 ^cd^	23.0 ^bc^	28.8 ^b^	27.4 ^b^	62.6 ^a^
trans-2-undecenal	2.34 ^e^	2.19 ^e^	2.22 ^e^	2.35 ^e^	28.4 ^a^	22.2 ^b^	8.17 ^d^	15.9 ^c^
trimethylpyrazine	1.90 ^e^	9.09 ^de^	12.1 ^de^	18.3 ^cd^	25.7 ^c^	39.4 ^b^	41.2 ^b^	101 ^a^

## Data Availability

The data presented in this study are available on request from the corresponding author.

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
