# Peer review of "Effect of pH, Reducing Sugars, and Protein on Roasted Sunflower Seed Aroma Volatiles"

_foods, 2023, doi:10.3390/foods12224155_

Round 1
Reviewer 1 Report
Comments and Suggestions for Authors
After carefully reading the manuscript entitled: "Effect of pH, Reducing Sugars, and Protein on Roasted Sunflower Seed Aroma Volatiles", it can be concluded that the authors spent a lot of time and effort conducting experiments and writing an article. However, several things could be improved. Below are remarks and suggestions.
- The abstract should be restructured. Ensure the abstract contains essential information such as the research problem, methodology, significant findings, and concise conclusions. Please explain why the study is relevant and how it contributes to the field.
- The introduction offers a general background on sunflower seeds as a popular snack and touches upon the importance of aroma volatiles generated through the Maillard and lipid oxidation reactions. However, the depth and breadth of the background information are insufficient to contextualize the study thoroughly. The introduction could benefit from:
- More comprehensive references about the Maillard reaction and lipid oxidation in the context of food science,
- Information about previous work on sunflower seeds.
- The cited references are generally relevant, but the paper lacks citations that directly mention sunflower seeds and their aroma volatiles. More targeted references would strengthen the paper's foundation.
- The research design is methodologically sound. However, understanding the effects of added proteins on the formation of the Volatiles and the chemical composition of whey concentrate and whey isolate is necessary. It is unclear how iron and copper concentration was determined.
- It is unclear why in Sensory Analysis treatments consisting of kernels soaked at pH 4, pH 9 and 20% glucose are omitted.
- Applying high temperatures (above 120°C) to foods, including proteins and carbohydrates, causes some toxic compounds such as acrylamide. Determination of acrylamide is missing.
- The results are presented in a structured manner, with tables and graphs to support the findings. Statistical analyses are appropriately conducted to validate the results. However, some figures are not clearly labelled, affecting the readability. Furthermore, the potential toxicological effects of obtained products are missing.
- Discussion on the concentration of volatile organic compounds present in roasted sunflower seed and potential toxicological effects needs to be included.
- The quality of the English language is generally good but could benefit from further proofreading for minor grammatical and spelling mistakes.
- The manuscript shows promise but needs revisions for clarity, depth, and comprehensive referencing to meet the standards of academic excellence.
Comments on the Quality of English Language
- The quality of the English language is generally good but could benefit from further proofreading for minor grammatical and spelling mistakes.
Author Response
After carefully reading the manuscript entitled: "Effect of pH, Reducing Sugars, and Protein on Roasted Sunflower Seed Aroma Volatiles", it can be concluded that the authors spent a lot of time and effort conducting experiments and writing an article. However, several things could be improved. Below are remarks and suggestions.
- The abstract should be restructured. Ensure the abstract contains essential information such as the research problem, methodology, significant findings, and concise conclusions. Please explain why the study is relevant and how it contributes to the field.
The abstract has been expanded with additional information and its relevance explained.
- The introduction offers a general background on sunflower seeds as a popular snack and touches upon the importance of aroma volatiles generated through the Maillard and lipid oxidation reactions. However, the depth and breadth of the background information are insufficient to contextualize the study thoroughly. The introduction could benefit from:
- More comprehensive references about the Maillard reaction and lipid oxidation in the context of food science,
Additional discussion and references concerning Maillard and lipid oxidation have been added into the introduction.
- Information about previous work on sunflower seeds.
Additional discussion and references on sunflower seeds has been added in the introduction.
- The cited references are generally relevant, but the paper lacks citations that directly mention sunflower seeds and their aroma volatiles. More targeted references would strengthen the paper's foundation.
More discussion of sunflower seeds and their volatiles has been added in the introduction.
- The research design is methodologically sound. However, understanding the effects of added proteins on the formation of the Volatiles and the chemical composition of whey concentrate and whey isolate is necessary. It is unclear how iron and copper concentration was determined.
A reference is included to clarify this.
- It is unclear why in Sensory Analysis treatments consisting of kernels soaked at pH 4, pH 9 and 20% glucose are omitted.
An explanation on why those samples were chosen has been added to the discussion.
- Applying high temperatures (above 120°C) to foods, including proteins and carbohydrates, causes some toxic compounds such as acrylamide. Determination of acrylamide is missing.
Measurement of acrylamide is beyond the scope of this project. See the literature for this information. For instance:
Asadi S, Aalami M, Shoeibi S, Kashaninejad M, Ghorbani M, Delavar M. Effects of different roasting methods on formation of acrylamide in pistachio. Food Sci Nutr. 2020 May 12;8(6):2875-2881. doi: 10.1002/fsn3.1588. PMID: 32566205; PMCID: PMC7300066.
Süvari M., Sivri G., Öksüz Ö. (2017). Effect of different roasting temperatures on acrylamide formation of some different nuts. Journal of Environmental Science, Toxicology and Food Technology, 11(4), 38-43 https://doi.org/10.9790/2402-1104013843
Nematollahi A., Kamankesh M., Hosseini H., Hadian Z., Ghasemi J., Mohammadi A. (2020). Investigation and determination of acrylamide in 24 types of roasted nuts and seeds using microextraction method coupled with gas chromatography-mass spectrometry: central composite design. Journal of Food Measurement and Characterization, 1-13 https://doi.org/10.1007/s11694-020-00373-9
Becalski A., Lau B., Lewis D., Seaman S. (2003). Acrylamide in Foods: Occurrence, Sources, and Modeling. Journal of Agriculture and Food Chemistry, 51, 802-808 https://doi.org/10.1021/jf020889y
- The results are presented in a structured manner, with tables and graphs to support the findings. Statistical analyses are appropriately conducted to validate the results. However, some figures are not clearly labelled, affecting the readability. Furthermore, the potential toxicological effects of obtained products are missing.
Some of the labels were rewritten and one figure was replaced. Toxicology is beyond the scope of this study.
- Discussion on the concentration of volatile organic compounds present in roasted sunflower seed and potential toxicological effects needs to be included.
The formation of the volatiles is discussed in the results section. Toxicology is beyond the scope of this study.
- The quality of the English language is generally good but could benefit from further proofreading for minor grammatical and spelling mistakes.
The grammatical and spelling mistakes have been corrected
- The manuscript shows promise but needs revisions for clarity, depth, and comprehensive referencing to meet the standards of academic excellence.
The manuscript has been revised and additional references added.
Comments on the Quality of English Language
- The quality of the English language is generally good but could benefit from further proofreading for minor grammatical and spelling mistakes.
The grammatical and spelling mistakes have been corrected
Reviewer 2 Report
Comments and Suggestions for Authors
It is well presented and interested piece of research and I support the paper.
My only advice is that such analysis should be also considered in frames of highly independent and objective physical methods, such as gas chromatography, maybe in a modern version with the supplementary electric field-based separation. It is not the only such method. The usage could support and extend conclusions. Maybe these issues should be a little discussed in one of the sections.
This is the reason I can not indicate the top, 'high' merit. Nevertheless, I recommend it.
Author Response
It is well presented and interested piece of research and I support the paper.
My only advice is that such analysis should be also considered in frames of highly independent and objective physical methods, such as gas chromatography, maybe in a modern version with the supplementary electric field-based separation. It is not the only such method. The usage could support and extend conclusions. Maybe these issues should be a little discussed in one of the sections.
This is the reason I can not indicate the top, 'high' merit. Nevertheless, I recommend it.
The explanation of how the analysis was performed, has been expanded.
Reviewer 3 Report
Comments and Suggestions for Authors
The paper describes the effect of the presence of some nutritional ingredients of the sunflower seed on it's aroma compounds. In this article authors tried to show how pH, reducing sugars, and proteins can change the aroma compounds.
This article does not present how the content of volatile compounds was calculated. It is unclear how the quantitative analysis determining of the volatiles was performed? How authors could present in the Table 2 the amounts of the volatile compounds in ppb? Please explain that.
How the sensory analysis was performed? Did the authors have consent from relevant ethical committees to conduct research on humans? Please add some information about it.
In the paragraph materials and methods please explain how the measurement of pH was performed?
Moreover, please add methods for quantification of the content of reducing sugars, proteins, measurement of the color.
Figure No. 3 is illegible, please change it and add the values on the chart along with statistical analysis.
Why authors were sure which kind of volatile compounds are desirable in the aroma of the sunflower seed?
What does it mean that the sample was most liked? on what basis authors can be sure that isolate and whey concentrate was the most liked samples?
Why authors did no show the aroma compound concentration of whey protein and other products used as addition to the sunflower seed.
Discussion of the results is inappropriate, it does not apply to similar results of analyzes of sunflower seeds.
Conclusion is too long and should be based only on the given results in the paper.
Author Response
The paper describes the effect of the presence of some nutritional ingredients of the sunflower seed on it's aroma compounds. In this article authors tried to show how pH, reducing sugars, and proteins can change the aroma compounds.
This article does not present how the content of volatile compounds was calculated. It is unclear how the quantitative analysis determining of the volatiles was performed? How authors could present in the Table 2 the amounts of the volatile compounds in ppb? Please explain that.
The calculation of the volatile compounds is now explained more completely in the Methods section.
How the sensory analysis was performed? Did the authors have consent from relevant ethical committees to conduct research on humans? Please add some information about it.
The section was revised. The sensory study was approved by the Ohio State University Institutional Review Board and the approval number is provided at the end of section 2.4.
In the paragraph materials and methods please explain how the measurement of pH was performed?
This has been added.
Moreover, please add methods for quantification of the content of reducing sugars, proteins, measurement of the color.
This section has been revised.
Figure No. 3 is illegible, please change it and add the values on the chart along with statistical analysis.
Figure 3 has been replaced.
Why authors were sure which kind of volatile compounds are desirable in the aroma of the sunflower seed?
This is based on a review of previously published work. The introduction was revised to be more clear.
What does it mean that the sample was most liked? on what basis authors can be sure that isolate and whey concentrate was the most liked samples?
These results were determined by sensory analysis, and statistical calculation. This section has been expanded.
Why authors did no show the aroma compound concentration of whey protein and other products used as addition to the sunflower seed.
The treatments had negligible aroma concentrations, so the results were not included.
Discussion of the results is inappropriate, it does not apply to similar results of analyzes of sunflower seeds.
Sections of the discussion were revised.
Conclusion is too long and should be based only on the given results in the paper.
The conclusion has been revised.
Round 2
Reviewer 1 Report
Comments and Suggestions for Authors
Revised paper can be accepted for publication.
Author Response
Thank you
Reviewer 3 Report
Comments and Suggestions for Authors
In the paragraph materials and methods please explain how pH-meter was prepared before measurement?
Please add methods for quantification of the content of reducing sugars and proteins. Which kid of methods were used to determine the sugar and protein content in the sample?
Why authors did no show the aroma compound concentration of whey protein and other products used as addition to the sunflower seed.
Even though, The treatments had negligible aroma concentrations, authors should mentioned about it, if the analysis were not performed the experiment is incomplete.
I think that the discussion of the results are not enough and also the conclusion section was not changed, as I asked in the last revision. “Discussion of the results is inappropriate, it does not apply to similar results of analyzes of sunflower seeds. Conclusion is too long and should be based only on the given results in the paper. “
Author Response
In the paragraph materials and methods please explain how pH-meter was prepared before measurement?
-Added
Please add methods for quantification of the content of reducing sugars and proteins. Which kid of methods were used to determine the sugar and protein content in the sample?
- added
Why authors did no show the aroma compound concentration of whey protein and other products used as addition to the sunflower seed. Even though, The treatments had negligible aroma concentrations, authors should mentioned about it, if the analysis were not performed the experiment is incomplete.
-A table of these values was added to the appendix, and a discussion of the volatile levels in the treatments at the beginning of the results section.
I think that the discussion of the results are not enough and also the conclusion section was not changed, as I asked in the last revision. “Discussion of the results is inappropriate, it does not apply to similar results of analyzes of sunflower seeds. Conclusion is too long and should be based only on the given results in the paper. “
-Additional discussion was added. The conclusion was shortened by removing the implications and practical significance of the work